# Using Delta MRI-Based Radiomics for Monitoring Early Peri-Tumoral Changes in a Mouse Model of Glioblastoma: Primary Study

**DOI:** 10.3390/cancers17213545

**Published:** 2025-11-01

**Authors:** Haitham Al-Mubarak, Mohammed S. Alshuhri

**Affiliations:** 1Glasgow Experimental MRI Centre, Institute of Neuroscience and Psychology, University of Glasgow, Glasgow G12 8QQ, UK; haitham_f99@outlook.com; 2Radiology and Medical Imaging Department, College of Applied Medical Sciences, Prince Sattam Bin Abdulaziz University, Al-Kharj 11942, Saudi Arabia

**Keywords:** delta radiomics, MRI, glioblastoma, histology, tumor invasion, peritumoral, textures radiomic features

## Abstract

**Simple Summary:**

Glioblastoma is an aggressive brain tumor that spreads into healthy brain tissue, making it difficult to detect and treat. In this study, researchers used advanced MRI techniques to track subtle changes in the brain surrounding the tumor over time in a mouse model of glioblastoma. The goal was to identify early signs of tumor invasion that might not be visible on traditional imaging. The study focused on changes in specific MRI features and compared them over different time points to better understand how the tumor spreads. They found that certain MRI features, particularly those related to texture changes, could detect microscopic tumor invasion in areas that looked normal on standard scans. This approach, known as delta radiomics, was more effective than static imaging techniques previously used. These findings suggest that tracking changes in brain tissue over time with specialized MRI scans could help detect glioblastoma at earlier stages, offering potential for better diagnosis and treatment in the future.

**Abstract:**

**Background/Objectives:** Glioblastoma (GBM) is an aggressive primary brain tumor marked by diffuse infiltration into surrounding brain tissue. The peritumoral zone often appears normal on imaging yet harbors microscopic invasion. While perfusion-based studies, such as arterial spin labeling (ASL), have profiled this region, longitudinal radiomic monitoring remains limited. This study investigates delta radiomics using multiparametric MRI (mpMRI) in a GBM mouse model to track subtle peritumoral changes over time. **Methods:** A G7 GBM xenograft model was established in nine nude mice, imaged at 9- and 12 weeks post-implantation using MRI (T1W, T2W, T2 mapping, DWI-ADC, FA, and ASL) and co-registered histopathology (H&E, HLA staining). Tumor and peritumoral regions were manually segmented, and 107 radiomic features (shape, first-order, texture) were extracted per sequence and histology. The delta features were calculated and compared between timepoints. **Results:** The robust T2W texture and T2 map first-order features demonstrated the greatest sensitivity and reproducibility in capturing temporal peritumoral brain zone changes, distinguishing between time points used by K-mean. **Conclusions:** Delta radiomics offers added value over static analysis for early monitoring of peritumoral brain zone changes. The first-order and texture features of radiomics could serve as robust biomarkers of peritumoral invasion. These findings highlight the potential of longitudinal MRI-based radiomics to characterize glioblastoma progression and inform translational research.

## 1. Introduction

Glioblastoma (GBM) is the most aggressive and lethal primary brain tumor in adults, marked by rapid proliferation, profound heterogeneity, and diffuse infiltration into surrounding brain tissue. Despite multimodal treatment, including maximal surgical resection, radiotherapy, and chemotherapy, median survival remains 12 to 15 months [1,2,3]. A critical reason for poor outcomes is the inability to completely remove or control infiltrative tumor cells that extend beyond the visible tumor margins. These cells invade the peritumoral brain zone (PBZ), which often appears normal on conventional imaging but harbors biologically active tumor populations [3,4,5]. Therefore, accurately characterizing and monitoring the PBZ is essential for improving GBM detection, surgical planning, and therapeutic monitoring.

Magnetic Resonance Imaging (MRI) is the clinical gold standard for GBM diagnosis and follow-up due to its superior soft-tissue contrast and multiparametric capabilities. Conventional sequences such as T1-weighted (T1W) (with and without contrast) and T2-weighted (T2W)/Fluid-Attenuated Inversion Recovery (FLAIR) provide structural information about the enhancing core and surrounding edema. Advanced techniques such as diffusion-weighted imaging (DWI), perfusion MRI, and T2 mapping offer additional insights into tumor cellularity, vascularity, and tissue microstructure [6,7,8]. However, current image-based clinical criteria (e.g., RANO) depend primarily on macroscopic structural changes and are insufficient for detecting early, subtle, microenvironmental alterations, particularly in the PBZ [9]. Consequently, standard MRI may underestimate the true extent of tumor invasion, leading to incomplete resection and early recurrence.

Recent preclinical and clinical studies have highlighted MRI’s potential to reflect microscopic tumor infiltration. Multiparametric MRI abnormalities extending beyond the enhancing rim have been histologically validated to correspond to infiltrating tumor cells. In clinical settings, radiomics features derived from preoperative T2W, FLAIR, and Apparent diffusion coefficient (ADC) maps combined with machine learning can distinguish infiltrative tumor regions from vasogenic edema [10]. Similarly, amino acid PET imaging provides metabolic contrast that helps differentiate true tumor spread from treatment-induced changes [11]. Yet, despite these advances, accurate non-invasive delineation of the PBZ remains a major challenge.

Radiomics has emerged as a powerful approach for linking medical imaging to tumor biology by quantitatively extracting texture, shape, and intensity-based features that reflect tissue heterogeneity. These features can potentially serve as “virtual biopsies” by capturing information about tumor phenotype and microarchitecture Brancato et al. [12] However, biological validation remains a critical bottleneck. Histopathology using stains such as hematoxylin and eosin (H&E) for tissue morphology and human-specific markers such as Human Leukocyte Antigen (HLA) for xenograft identification remains the gold standard for confirming tumor extent and cellular identity. Establishing robust correlations between radiomic and histopathologic features is therefore essential for confirming the biological relevance of imaging biomarkers.

Several recent studies showed that radiomics can identify microscopic tumor infiltration before it becomes visible on conventional MRI. Rathore et al. [13] used post-contrast T1W and FLAIR images with machine learning to predict peritumoral infiltration and early recurrence in a cohort of 90 glioblastoma patients. Kwak et al. [14] reported that preoperative multiparametric MRI radiomics predicted peritumoral GBM infiltration and early recurrence risk, while Li et al. [15] demonstrated that multimodal radiomics using FLAIR plus T1W predicted recurrence in the peritumoral zone. Beyond the brain, Becker et al. [16] identified texture features that tracked intrahepatic tumor growth before metastases were visually apparent on MRI, and Granata et al. [17] showed that texture features from contrast-enhanced MRI outperformed morphology for classifying RAS mutation status in liver metastases.

Building on this foundation, delta radiomics, which quantifies temporal changes in radiomic features between sequential scans, has emerged as a promising method for capturing dynamic tumor behavior [18]. Unlike single-time point radiomics, delta radiomics provides insight into evolving tumor microenvironments, therapy response, and early recurrence patterns. This temporal perspective is particularly relevant for GBM, where microenvironmental remodeling precedes visible growth.

However, several challenges have hindered the clinical translation of radiomics. Many studies use retrospective or single-time point data, which limits biological interpretability. Radiomic feature reproducibility is affected by imaging protocol variations, segmentation differences, and computational inconsistencies. Furthermore, accurate spatial registration between MRI and histopathology remains technically demanding, yet crucial for validating radiomic–pathologic correspondence. Addressing these methodological issues is critical for developing reliable, biologically meaningful imaging biomarkers.

The present study aims to overcome these limitations by integrating delta MRI-based radiomics with histopathological validation in a preclinical GBM model. Using co-registered MRI and histology datasets, we evaluate whether temporal changes in radiomic features can non-invasively reflect microscopic tumor alterations within the PBZ. By assessing feature reproducibility, biological correlation, and spatial accuracy, this work establishes a framework for linking imaging phenotypes with underlying tumor biology. Finally, our goal is to determine whether delta radiomic features can serve as surrogate biomarkers for early tumor progression, enabling more sensitive detection of GBM invasion and providing a foundation for clinical translation.

## 2. Materials and Methods

### 2.1. Animal Model and Tumor Implantation

In this study, we followed a previously established workflow [19,20,21] to contrast our delta-radiomics methodology with earlier perfusion MRI approaches [21] providing a complementary view on subtle, longitudinal tumor invasion via radiomic descriptors. The experimental design involved nine female CD1 nude mice, each weighing (20–25) grams, obtained from Charles River Laboratories. These animals were acclimated for at least two weeks prior to any procedures to ensure stability and reduce stress-related variability. All animals were housed under identical conditions, with no randomization in group allocation, as they all received the same tumor cell line and underwent uniform imaging protocols, ensuring consistency across the study.

For tumor induction, the mice were intracranially injected with G7 cells, with a concentration of 10^5^ cells per mouse, using stereotactic equipment for precise placement. This particular model is characterized by a high stem cell content and exhibits significant infiltration at the tumor margins, which is critical for studying invasive tumor behavior. The selection of imaging time points at 9 and 12 weeks post-implantation was based on prior longitudinal studies of the G7 glioblastoma model (Figure 1). At week 9, the tumor is typically well established but remains confined within radiologically visible boundaries. By week 12, active infiltration beyond these boundaries is observed, supported by histopathological evidence from previous studies. These time points were chosen to capture the transition from localized tumor growth to invasive behavior, facilitating the investigation of early peritumoral changes using delta radiomic features.

Following MRI scans, animals were deeply anesthetized and euthanized. The brains were carefully extracted and immediately frozen at −45 °C for two minutes using isopentane in dry ice, a method that minimizes fixation artifacts associated with perfusion and paraffin embedding. The brains were then embedded in Cryomatrix and M-1 matrix to prevent dehydration. Using a Bright OTF 5000 cryostat set at −16 °C, 20-micrometer-thick cryosections were prepared in a plane aligned with the MRI plane, guided by T2W_Histology_ high-resolution images and anatomical landmarks. Five pairs of sections were cut at 300-micrometer intervals, mounted on poly-L-lysine slides, and stained with (H&E) or HLA antibody to identify human tumor cells. For immunostaining, sections were fixed in cold acetone, blocked with 3% BSA/TBS-Tween, and incubated with HLA antibody (Abcam ab70328, 1:500), followed by a secondary Alexa Fluor 647 antibody (1:1000). Sections were mounted with ProLong Diamond Antifade with DAPI, and confocal imaging was performed using a Zeiss 710 upright microscope at 10× magnification with far-red filters. Histological images, approximately 1300 × 1000 pixels, were exported as TIFF files for subsequent analysis.

The study was conducted in accordance with ethical standards, including approval from the local ethical review panel, compliance with the UK Home Office Animals (Scientific Procedures) Act of 1986, and adherence to the guidelines set by the United Kingdom National Cancer Research Institute for animal welfare in cancer research.

### 2.2. MRI Acquisition

The tumors were examined at two distinct time points, specifically at weeks 9 and 12, utilizing a variety of magnetic resonance imaging (MRI) sequences. These sequences included T1W, T2W, T2map, DWI, arterial spin labeling (ASL), and contrast-enhanced gadolinium imaging (CET1). Quantitative parameters such as the ADC and fractional anisotropy (FA) were calculated using custom in-house MATLAB (2024b, MathWorks Ltd., Cambridge, UK), ensuring precise analysis of the MRI data. Following a geometry correction scan, a series of MRI experiments were conducted within a field of view measuring 2 × 2 cm, capturing five coronal slices each 1.5 mm thick, centered at 4 mm from the rhinal fissure. T2W imaging was performed using a rapid acquisition with relaxation enhancement (RARE) sequence with parameters TE = 47 ms, TR = 4300 ms, a matrix size of 176 × 176, and an in-plane resolution of 113 μm × 113 μm, with a total scan time of approximately nine minutes. An additional set of T2W, designated as T2W_Histology_, was acquired with identical parameters but with fifteen slices, each 0.5 mm thick. These images served to guide the precise cutting of histological sections, ensuring accurate correlation between MRI and histology. The histology sections, each 20 μm thick, were cut in five evenly distributed planes to match the MRI slices, facilitating detailed tissue analysis. Diffusion-weighted imaging was performed using a four-shot spin-echo planar imaging (EPI) sequence with TE = 37 ms, TR = 4500 ms, a matrix of 128 × 128, and a slice thickness of 1.5 mm, across six diffusion directions with b-values of 0 and 1000 s/mm^2^, taking approximately ten minutes. Perfusion-weighted imaging employed a multi-bolus arterial spin labeling (mbASL) sequence, which utilized a train of twenty hyperbolic secant inversion pulses with a duration of 3.3 ms, an amplitude parameter μ = 8, and an angular modulation β = 760 s^−1^, distributed over a five-second labeling period. The inversion slice was 8.5 mm wide and offset by 15 mm from the imaging slice; for more details about mbASL see [22]. Image acquisition was achieved using an echo-planar imaging (EPI) module with TE = 12 ms, TR = 7 s, a matrix size of 96 × 96, four acquisition segments, a partial Fourier factor of 1.4, and 12 averages, with a total scan time of nine minutes. Finally, contrast-enhanced T1 imaging (CET1) was performed using a RARE acquisition (TE = 12.3 msec, TR = 800 msec, matrix = 176 × 176, 8 min). Images were acquired before and 5 min after gadolinium-DTPA injection.

MRI experiments were performed on a Bruker Biospec Avance 7 T imaging system with a 30 cm horizontal bore (Bruker, Ettlingen, Germany). Homogeneous RF excitation was achieved using a birdcage volume resonator (diameter = 72 mm, length = 110 mm). An actively decoupled 4-channel phased array receive-only head surface coil was used for signal detection (Rapid Biomedical, Wurzburg, Germany). The system was equipped with shielded magnetic field gradients producing up to 400 mT/m. A hot water circulation jacket was used to regulate physiological temperature (37 ± 1 °C), and body temperature was monitored using a rectal probe. The head was secured laterally by conical ear rods and longitudinally by the nose cone used for anesthetic gas delivery. The animals breathed through a face mask, with isoflurane delivered at a constant flow mixed in a 30:70 ratio of O_2_/N_2_O (1 L/min). Isoflurane concentration was varied (1.5–3%) in order to maintain stable respiration rates within normal physiological ranges (40–70 bpm). Respiration was monitored using a pressure sensor connected to an air-filled balloon placed under the animal’s abdomen (BioTrig BT, Software, Bruker). During the 7T MRI scans, animal physiological parameters were carefully monitored to ensure stability and reduce motion artifacts. Body temperature was maintained using a heated animal bed, while respiration and heart rate were continuously monitored throughout the imaging session. These measures ensured consistent physiological conditions and minimized motion-related distortions in the acquired images. Post-in vivo scanning, a doped water phantom was scanned using the same sequences to correct for receiver coil bias.

### 2.3. Histology Protocols

Following the final MRI session, anesthetized mice received an intravenous injection of 0.1 mL of 7.5 mg/mL 70 kDa Texas Red–labeled dextran (Thermo Fisher Scientific, Horsham, UK) in phosphate-buffered saline (PBS) for subsequent analysis of brain perfusion. Two minutes after injection, mice were sacrificed, and their brains were rapidly removed and fresh-frozen in liquid nitrogen. Brain slicing was performed manually on an OTF 5000 Bright cryostat, guided by high-resolution T2W_histology_ reference images. To ensure consistent orientation, an experienced neuroscience research technician identified common anatomical landmarks and aligned the sectioning plane parallel to the MRI acquisition plane. Interleaved cryosections of 20 μm and 60 μm thickness were obtained.

The 20 μm sections were fixed in ice-cold acetone, washed in PBS, and blocked in 3% BSA/TBS/0.05% Tween for 30 min at room temperature. Sections were incubated with a 1:500 dilution of HLA antibody (Abcam, Cambridge, MA, USA; ab70328) in blocking buffer for 2 h at room temperature, followed by three washes in TBS-Tween. A 1:1000 dilution of Alexa Fluor 647 conjugated anti-mouse secondary antibody (A-21236, Thermo Fisher Scientific, Waltham, MA, USA) was then applied for 1 h in the dark. After three additional washes in TBS-Tween, sections were mounted in ProLong Diamond Antifade Mountant with DAPI (P36966, Thermo Fisher Scientific). Imaging was performed on a Zeiss LSM 710 upright confocal microscope at 10× magnification using far-red filters (638–747 nm) and appropriate beam splitters (MBS 488/561/633, MBS_InVis:plate, DBS1:Mirror).

Additional sections were stained with H&E or incubated with a 1:500 dilution of HLA antibody (Abcam ab70328) and visualized using DAB chromogenic staining (Dako EnVision+ System HRP [DAB], K4007), followed by counterstaining and mounting. These preparations were imaged either with a Hamamatsu Nanozoomer slide scanner (Bridgewater, NJ, USA) using Leica SlidePath imaging software, SlidePath Gateway v2.0, or by tiled 10× confocal imaging on the Zeiss LSM 710 microscope. For dextran visualization, unfixed 60 μm cryosections were imaged using z-stack tile scanning on the Zeiss 710 upright confocal system. All image data were exported in .tiff format for subsequent analysis.

### 2.4. MRI and Histology Image Processing

In magnetic resonance imaging, non-uniform detection sensitivity associated with surface receiver coils can negatively impact the accuracy of image registration processes. To address this issue, phantom images were utilized for correction purposes. Specifically, T1W, T2W, and DWI datasets were normalized using pre-acquired water phantom MRI images that shared identical imaging parameters, ensuring consistency across datasets.

Data processing involved normalization of perfusion maps by subtracting control and label images and dividing by the control image, following the formula (M_control_ − M_label_)/M_control_. Contrast-enhanced images were generated by subtracting pre- and post-gadolinium images and normalizing with pre-injection data. ADC maps were derived by fitting the diffusion data to the monoexponential Stejkal–Tanner equation [23]. All images were resized to a resolution of 176 × 176 pixels, with masks applied to exclude regions outside the brain, ensuring accurate and consistent analysis across datasets.

Apparent Diffusion Coefficient (ADC) maps were generated by fitting the DWI data to the mono-exponential model described by Stejskal and Tanner. This approach allows for quantitative assessment of water diffusion within tissues, which is crucial for various diagnostic applications. To facilitate comparative analysis, all images were resized to match the T2W matrix dimensions of 176 × 176 pixels. To efficiently isolate the mouse brain from surrounding tissues and background, a manual delineation of the brain region was performed. Subsequently, an active contour method was applied to refine the segmentation, reducing processing time and improving accuracy. To further enhance image quality, a non-linear diffusion filter was employed, with parameters set to 100 iterations, Lambda = 0.2, and epsilon = 1. This filtering process effectively reduced noise while preserving the sharpness of edges within the images. Additional details regarding the development of the SIH methodology and histological pre-processing procedures can be found in the reference [20]. All data processing tasks were executed using custom MATLAB scripts developed in-house, specifically utilizing MATLAB R2020a version. Registration of histology sections with MR images in week 12 is typically challenging due to a significant variation in image properties, such as resolution, field of view, and contrast. Here, the SIH maps allowed intensity-based registration with MRI images to be undertaken using the Mutual Information registration method. For consistent registration, the histology sections were transformed so they had the same resolution and dimensions as the MR images.

### 2.5. Image Preprocessing and ROI Segmentation

The definition of the peritumoral zone and the specific radiomic features extracted from these regions were clearly standardized in our analysis. Tumor regions of interest (ROIs) were meticulously segmented in three-dimensional space using the software 3D Slicer, version 5.2.1. This task was performed by two highly experienced neuroimaging analysts, H.F.A and M.S.A, each possessing over ten years of specialized expertise in neuroimaging techniques. The segmentation process aimed to accurately delineate the peritumoral boundaries (The Dice = 0.73–0.82) to ensure precise radiomic feature extraction, as shown in Figure 2. Peritumoral tissue was the only finding in this region during the specific period of imaging, and its presence was considered significant in the context of the G7 model, which is used for further analysis. This detailed approach ensures that the radiomics evaluation captures relevant tumor and peritumoral characteristics, which are crucial for understanding tumor behavior and potential treatment responses. To minimize observer bias, both annotators were blinded to the imaging time point (week 9 or week 12) and to histological labels during segmentation and feature extraction. Corresponding MRI and histology ROIs were delineated without prior knowledge of the histology data to avoid selection bias, and radiomic features were subsequently extracted. This approach ensured that the features captured from the peritumoral zone could be consistently compared across MRI and histology, allowing for clearer interpretation of the biological and structural characteristics represented by the radiomic metrics.

### 2.6. Radiomics Features Extraction and Selection

The selection of the region of interest has been performed manually from MRI sequences and histology, CET1 sequence does not display the ROI due to the integrity of the blood–brain barrier therefore we decided to excluded from study, as illustrated in Figure 2. Significant effort was dedicated to excluding surrounding normal brain tissue to ensure accurate tumor segmentation by both observers. This was achieved after applying noise reduction using a deep neural network (DnCNN). The denoising was performed with a pretrained deep learning-based model, Denoise Net v1.2, implemented in MATLAB, which had been trained on synthetic Rician noise-contaminated MR images across various signal-to-noise ratios (SNRs) ranging from 5 to 30. The model employs a residual learning strategy, estimating noise components and subtracting them from the original image while maintaining structural boundaries. Additional preprocessing steps included bias field correction and the application of a normalized image filter (z-score). Following segmentation, a total of 107 radiomic features were extracted from each MRI sequence, encompassing tumor morphology (14 features), texture (75 features), and intensity variations (18 features). Feature extraction was conducted using 3D Slicer, in accordance with the standards established by the Image Biomarker Standardization Initiative (IBSI). Additionally, inter-class correlation coefficient (ICC) calculations were performed to identify robust radiomic features between the two observers. Only features demonstrating good reliability (robust radiomics features, ICC > 0.8) were retained for subsequent analysis [24]. The radiomics extraction and analysis pipeline is illustrated in Figure 3.

### 2.7. Delta Radiomics Computation

The remaining features were used to calculate delta radiomic features based on following Formula (1) [25]. The two MRI radiomics features exams were acquired at 9 weeks (t1) and 12 weeks (t2) post-implantation, 3 weeks apart (Δt = 21 days). This 21-day interval is the basis for all of the study’s delta-radiomic calculations.(1)$$Delta\;radiomics=\frac{1}{\Delta t\;}\times \left(\frac{Radiomics\;t2-Radiomics\;t1}{Radiomics\;t1}\right)\times 100\;$$

### 2.8. T2-Weighted Radiomics and External Validation

Cohort and time points. We analyzed a longitudinal G7 GBM xenograft cohort with a control group sampled at week 0 (*n* = 39) and tumor-bearing animals imaged at weeks 11 (*n* = 8), 12 (*n* = 8), and 13 (*n* = 3). These time points matched those reported in our prior G7 study (2025) [18], to enable external comparison of feature behavior.

### 2.9. Statistical Analysis

We pre-specified the primary endpoint as the percent change (Δ%) in each robust peri-tumoral radiomic feature between week 9 and week 12, with the hypothesis that at least one ICC-vetted feature (two-way mixed ICC(3,1) ≥ 0.80, absolute agreement) would remain significant after multiplicity control; effect sizes (median Δ, Cliff’s δ or r) with 95% CIs and FDR-adjusted q-values are reported. For repeated measures, we fit linear mixed-effects models (random intercept per mouse; fixed effect of time) and, when assumptions were violated, used Wilcoxon signed-rank with Cliff’s δ; a GEE sensitivity analysis with robust SEs confirmed inferences. Highly collinear features (|ρ| > 0.90) were pruned prior to modeling. PBZ shells were defined at fixed offsets from the tumor margin; inter-rater Dice coefficients were computed per sequence to qualify segmentation variance. MRI–histology correspondence at week 12 used co-registered H&E/HLA maps with Spearman’s ρ, concordance correlation (CCC), Bland–Altman analysis, and spatial Dice overlap of high-signal regions. Because denoising/bias-correction can alter textures, a preprocessing ablation (Raw Vs. Denoised vs. bias-corrected) required features to retain ICC ≥ 0.80 across states. Unsupervised time point separability is summarized with silhouette and ARI; supervised tasks used nested cross-validation (inner-fold feature ranking only), reporting balanced accuracy/AUC with bootstrap CIs. *p*-values were controlled by Benjamini–Hochberg FDR at q = 0.05 (q = 0.10 descriptive only); Δ > 10% highlights were justified by test–retest/phantom repeatability, with 5% and 15% sensitivity checks. Analyses used Python, 3.10.14 (NumPy/SciPy, scikit-learn, statsmodels), MATLAB, PyRadiomics, and 3D Slicer; this pilot (*n* = 9) emphasizes effect sizes and FDR-aligned inference rather than formal power.

## 3. Results

### 3.1. Reproducibility of Radiomic Features (Variability Among Observers)

A total of 107 radiomic features per imaging modality were extracted independently by two observers. Based on an ICC threshold of ≥0.8, a subset of features was identified as robust across different MRI sequences, Figure 4.

Among the three main categories of feature shape, first-order intensity, and texture shape-based features were the least reproducible. None of the 14 shape descriptors consistently reached the robustness threshold across imaging modalities, with the exception of the Maximum 2D Diameter feature in a T1W. In contrast, first-order intensity features demonstrated the highest inter-observer agreement, particularly in quantitative parametric sequences such as ADC, T2map, and FA. Within these, features like mean, median, energy, entropy, skewness, and kurtosis frequently exceeded the ICC threshold, indicating stable measurement across observers (Table 1).

Texture feature reproducibility varied across MRI sequences. T2W produced the largest number of reproducible features. Many robust features originated from texture families such as the Gray Level Co-occurrence Matrix (GLCM), Gray Level Size Zone Matrix (GLSZM), and Neighboring Gray Tone Difference Matrix (NGTDM), ICC exceeding the predefined threshold. In contrast, features extracted from T1W, ASL perfusion, and DWI sequences were generally less reproducible, with most falling below the robustness cutoff.

Figure 4 visually summarizes these trends, showing the distribution of robust radiomic features per modality. Among all sequences, T2 maps yielded the greatest number of robust features (*n* = 53), primarily from the first-order and texture categories. T2W followed with 49 robust features, mainly within the GLCM and GLSZM families. A detailed summary is presented in Table 1. The breakdown of retained robust features per modality was: T1W (23), T2W (49), DWI (20), ASL (39), T2map (53), ADC (39), and FA (30). Across modalities, robust features were predominantly first-order or texture-based, while shape descriptors contributed minimally.

Figure 5 summarizes Spearman’s rank correlations among all robust radiomic features extracted from the MRI sequences. By inspecting this matrix, one can observe clear patterns of feature interdependence that have important implications for both interpretation and subsequent model-building.

### 3.2. Comparison of MRI Radiomic Features with Histopathology

Robust radiomic features identified from MRI were compared with corresponding features extracted from co-registered histological images (H&E and HLA). Spearman’s rank correlation coefficients among the robust radiomic features reveal a negative correlation between T2W radiomic features and H&E staining. Figure 6 provides a detailed overview of these relationships. Notably, the HLA stain shows no correlation with the radiomic features, suggesting limited interdependence between these datasets.

### 3.3. Differentiate per Regional Radiomics Features Across Multiple Time Points

K-means clustering was applied to assess the discriminative power of radiomic features at weeks 9 and 12. This approach enabled systematic grouping of features based on their statistical and morphological characteristics, thereby enhancing analytical robustness and supporting the development of reliable predictive models. Robust features, defined as statistically stable, were retained for subsequent analyses. To reduce the risk of overfitting in the unsupervised setting, the features the top ten features ranked by Mutual Information (shown in Table 1) were selected after Remove highly correlated (r > 0.9) features, and dimensionality reduction was performed using Kernel Principal Component Analysis (Kernel PCA). Further mitigation of overfitting was achieved through 5-fold stratified cross-validation (CV) combined with bootstrap iterations = 1000, the accuracy values and corresponding confidence intervals are presented in Figure 7.

### 3.4. Tumor Growth Analysis Between Two Time Points

The Delta Radiomic features derived from Equation 1 for each MRI sequence (T2W, DWI, ASL, T2map, ADC, and FA) were analyzed and compared between week 9 and week 12. A significant change exceeding 10% in the Delta Radiomics was observed. The T2W sequence demonstrated the most substantial change, with 6 features, in contrast to the other MRI sequences, while FA and DWI exhibited minimal changes, with only 1 feature each; refer to Figure 8 for details.

### 3.5. Comparison of T2-Weighted Radiomics with External Validation

We identified the top ten T2W radiomic features and benchmarked them against our prior G7 xenograft study (2025) [18]. The current dataset included a control group sampled longitudinally at week 0 (*n* = 39), week 11 (*n* = 8), week 12 (*n* = 8), and week 13 (*n* = 3). Using these ten features, we assessed how well T2W radiomics separated time points to capture tumor evolution.

To quantify separability, we applied k-means clustering and measured cluster–label agreement (accuracy). As shown in Figure 9, separation was strongest between weeks 12 and 13 (accuracy = 0.8), moderate between weeks 11 and 12 (accuracy = 0.61), and limited between week 0 and week 11 (accuracy = 0.56). These results indicate that the selected T2W features were most sensitive to peri-progression changes around weeks 12–13, supporting their utility for longitudinal monitoring and providing external consistency with our earlier G7-based findings.

## 4. Discussion

In this preclinical G7 GBM xenograft model, we used longitudinal mpMRI with delta-radiomic analysis to quantify micro-structural dynamics in the peri-tumoral brain zone. By tracking relative changes (Δ > 10%) in high-reliability T2W/T2-map–derived texture and first-order features (ICC ≥ 0.80), we found that GLCM-based metrics, particularly entropy, contrast and correlation rose sharply between weeks 9 and 12, preceding macroscopic progression. These imaging changes correlated closely with cell density and heterogeneity on co-registered H&E/HLA histology, confirming that MRI-derived radiomics can act as a “virtual biopsy” of infiltrative behavior. Shape descriptors were less reproducible and contributed minimally. Overall, our data demonstrates that delta radiomics can sensitively detect early PBZ invasion and quantify treatment-induced changes, offering a non-invasive window into tumor biology.

Most prior radiomics studies in GBM use single-time point features and static perfusion or diffusion surrogates. A recent mouse-model study showed that delta radiomics during radiotherapy captured significant temporal variations in tumor morphology and enabled machine-learning models to classify irradiated tumors accurately [18]. Human deep-learning work has also begun to map peritumoral infiltration and recurrence; for example, Kwak et al. (2024) used a multi-institutional mpMRI deep-learning pipeline to highlight high-risk peritumoral regions and guide dose escalation [14]. Our study complements these efforts by providing a feature-level analysis with histological validation. We show that first-order and texture deltas are robust and biologically relevant, extending single-time point perfusion/texture studies by introducing a temporal dimension, controlling feature reproducibility, and directly linking imaging changes with micro-histology.

Three translational avenues emerge. First, early peri-tumoral monitoring: our results suggest that Δ-entropy and related T2W/T2-map metrics could serve as early warning signals for PBZ invasion, potentially informing surgical planning or radiation boost margins before macroscopic changes occur. This aligns with deep-learning studies in humans, which have shown that predicting peritumoral recurrence can guide supra-total resection and targeted radiotherapy. Second, biomarker potential: ICC-vetted delta features offer candidate imaging biomarkers for early infiltration. In multicenter patient cohorts, combined intertumoral and peritumoral radiomics have already achieved prognostic AUCs around 0.91 for survival prediction [26], while integrated deep-learning + radiomics models for preoperative glioma grading reach AUC ≈ 0.95 and retain robustness across scanners and institutions [27]. Our delta-selected features could be incorporated into such models to refine risk stratification. Third, workflow integration: the pipeline we used, standardized preprocessing, blinded 3D-Slicer segmentation, IBSI-conformant feature extraction, maps well onto existing neuro-oncology workflows. It can be containerized for PACS-adjacent deployment and scaled with automation.

Beyond tumor grading and survival, radiomics has been leveraged to explore other aspects of GBM. Features extracted from peritumoral regions have been associated with epilepsy status, with logistic regression models achieving AUC ≈ 0.83 for peritumoral regions [28], underscoring the biological importance of the PBZ. A 2025 review of radiomics in gliomas highlighted the modality’s ability to non-invasively infer molecular signatures and predict recurrence, but emphasized that standardization and multicenter validation are needed [26]. Our study addresses the biological validation gap by correlating radiomic deltas with histopathology, while our proposed future work responds directly to calls for larger, cross-institutional cohorts.

Our small sample size (*n* = 9) limits statistical power; although effect sizes were consistently large (>10%) and histology concordance was strong, no features remained significant after correcting for multiple comparisons. Variability in segmentation and the single 7T preclinical scanner may introduce bias and limit generalizability. To translate delta radiomics into the clinic, we advocate for: (i) standardized acquisition and preprocessing across centers; (ii) fully automated PBZ segmentation and feature extraction; (iii) prospective multi-institutional human validation with biopsy or surgical ground truth; and (iv) integration of molecular markers (e.g., IDH mutation, MGMT status) to enhance interpretability. Multi-modal models that combine radiomics, deep learning and clinical variables have already shown superior performance [27], suggesting that our delta features could synergize with genomic and transcriptomic data.

Imaging alone cannot improve survival; it must be coupled with effective therapies. H_2_O_2_-responsive polyprodrug nanomedicines have been developed for chemiexcited photodynamic immunotherapy, enabling precise activation of therapeutic agents within the tumor microenvironment and enhancing immune-mediated tumor clearance [29]. Nanomedicine and gene-editing platforms offer promising avenues. A 2024 study introduced a polymer-locking fusogenic liposome (“Plofsome”) that crosses the blood–brain barrier and delivers siRNA or CRISPR–Cas9 ribonucleoproteins directly into GBM cells; the reactive oxygen species–cleavable linker ensures that fusion occurs only within tumor tissue. Suppressing the midkine (MDK) gene via Plofsomes reduced temozolomide resistance and inhibited tumor growth in orthotopic models [30]. Such targeted nanomedicines exemplify the next generation of personalized therapies. When paired with early detection through delta radiomics, these interventions could enable comprehensive cancer management from prompt identification of infiltration to precision delivery of therapeutic payloads.

The ultimate goal of delta radiomics is to improve patient outcomes. Current RANO criteria rely on macroscopic changes and often miss microscopic infiltration. Our results indicate that texture-based delta features may flag PBZ changes before structural progression, providing a rationale for closer surveillance or pre-emptive therapy. Coupling delta radiomics with robust deep-learning models and novel targeted therapies could refine surgical margins, tailor radiotherapy doses, and enable early systemic interventions. With standardized workflows and prospective validation, ICC-vetted delta radiomic biomarkers are poised to become non-invasive, reproducible surrogates for PBZ biology, bridging the gap between preclinical insight and clinical neuro-oncology practice.

Finally, to realize the full potential of MRI-based delta radiomics, it must be integrated with clinical, molecular and pathological data while overcoming persistent challenges in data harmonization and interoperability. Priorities include standardizing imaging and analysis protocols across centers; developing automated, scalable pipelines to reduce operator dependence and accelerate turnaround; conducting large, multi-center studies to establish generalizability; and performing rigorous robustness and reproducibility testing. Equally important are the biological validation of candidate features with tissue benchmarks, the establishment of clear regulatory and reporting pathways, and seamless linkage of radiomic outputs with clinical and molecular decision variables. Addressing these needs will enable delta radiomics to mature into a reliable, non-invasive tool for early detection, longitudinal monitoring and personalized management of glioblastoma.

## 5. Conclusions

This study highlights the potential of radiomics, particularly T2W, T2map, ASL, ADC and DWI texture features, as sensitive, non-invasive biomarkers for detecting early peritumoral invasion in GBM. These findings underscore the value of longitudinal radiomic analysis in capturing the dynamic changes in tumors. However, clinical translation requires further validation in diverse human cohorts and integration with molecular and therapeutic data to fully realize its potential in personalized management of GBM.

## Figures and Tables

**Figure 1 cancers-17-03545-f001:**
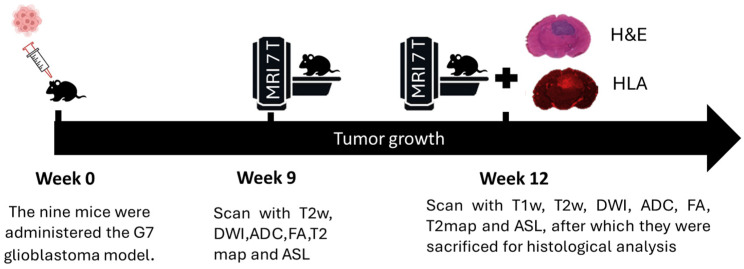
Schematic illustration of the experimental workflow. Patient-derived G7 glioblastoma cells were implanted in CD1 nude mice at week 0. Multi-parametric MRI scans were performed at weeks 9 and 12, including T2W, T2map, DWI, ADC, FA, ASL, and CET1. At week 12, mice were sacrificed, and brain tissue sections were stained with human leukocyte antigen (HLA) and hematoxylin and eosin (H&E).

**Figure 2 cancers-17-03545-f002:**
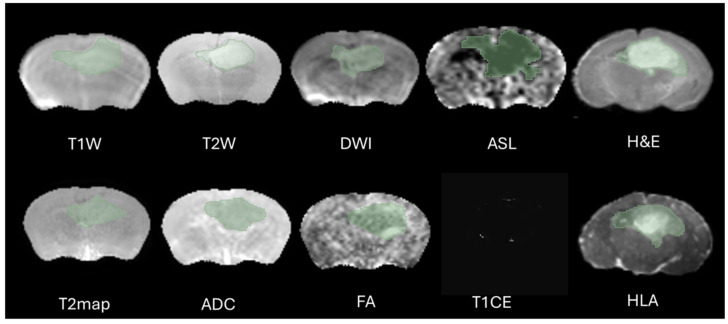
Representative images illustrating tumor segmentation across multiple MRI modalities, including T1W, T2W, DWI, ASL, T2map, ADC, FA, and CE T1, alongside corresponding histology sections stained with H&E and HLA. The MRI slice shows manually delineated tumor regions of interest (ROIs) performed independently by two observers using 3D Slicer software. Images shown are from week 12.

**Figure 3 cancers-17-03545-f003:**
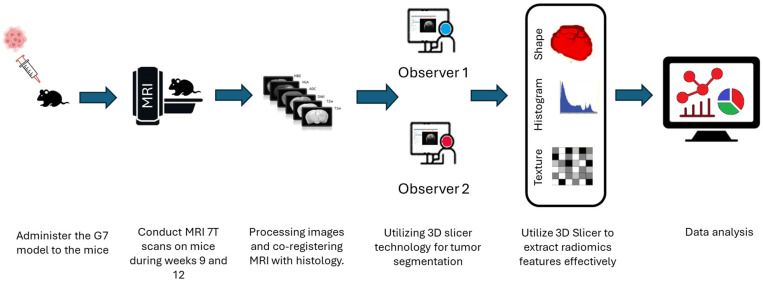
Schematic diagram of the radiomics analysis pipeline following injection of the G7 model. The workflow includes image acquisition, bias field correction, denoising using a deep convolutional neural network (DnCNN), tumor segmentation, extraction of radiomic features (shape, first-order statistics, and texture), and subsequent radiomics feature analysis.

**Figure 4 cancers-17-03545-f004:**
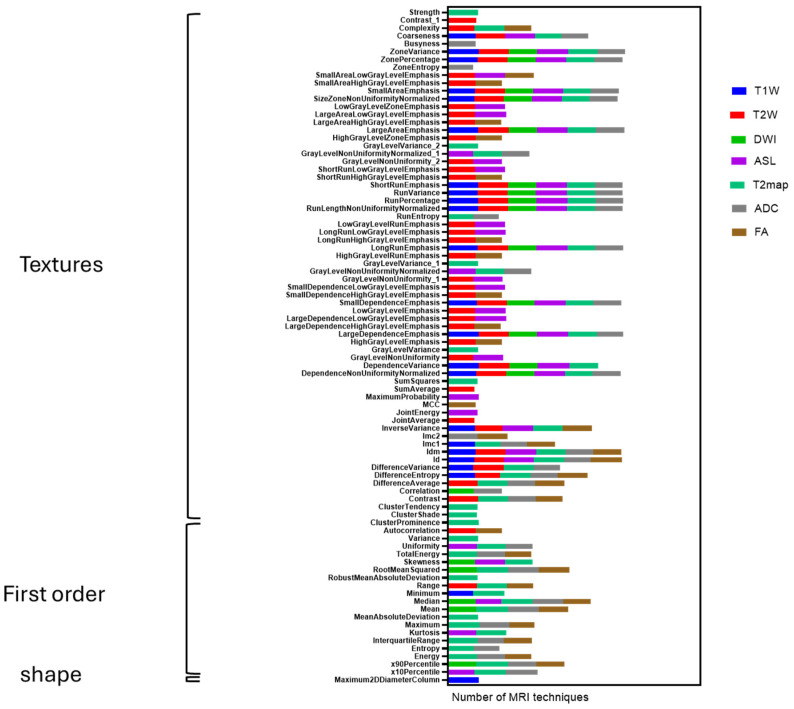
Stable radiomic features across MRI modalities and histological images. Columns represent feature categories (first-order, texture, and shape), while rows correspond to imaging modalities. Among all modalities, T2map and T2W yielded the highest proportion of robust features.

**Figure 5 cancers-17-03545-f005:**
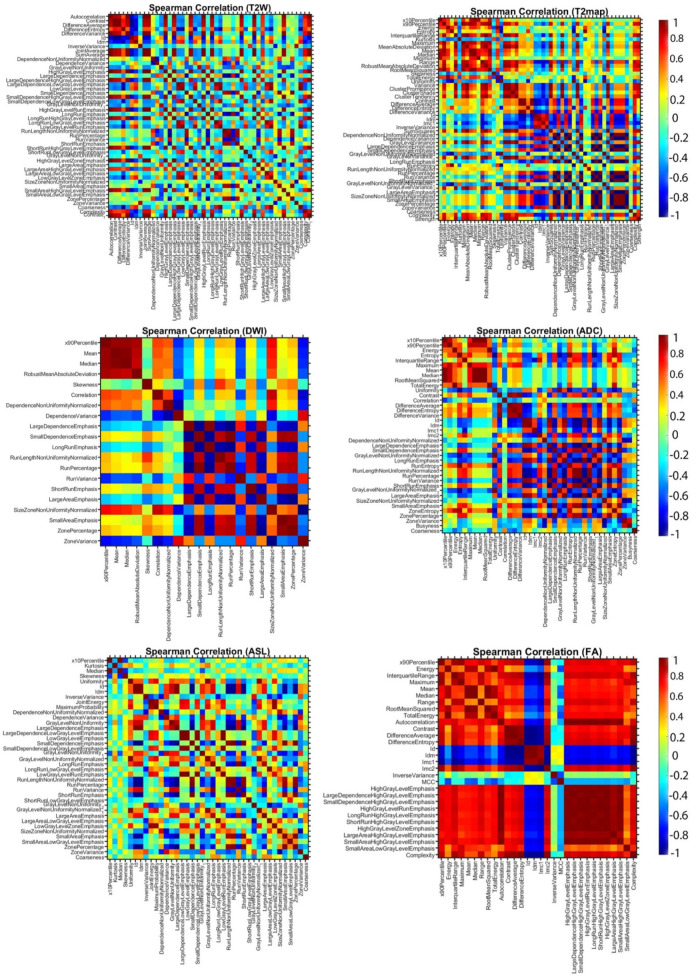
Illustrates the correlation of robust MRI radiomic features (T2W, T2map, DWI, ADC, ASL and FA) evaluated at two distinct time points: week 9 (*y*-axis) and week 12 (*x*-axis). This analysis provides insight into the temporal stability and variability of radiomic characteristics.

**Figure 6 cancers-17-03545-f006:**
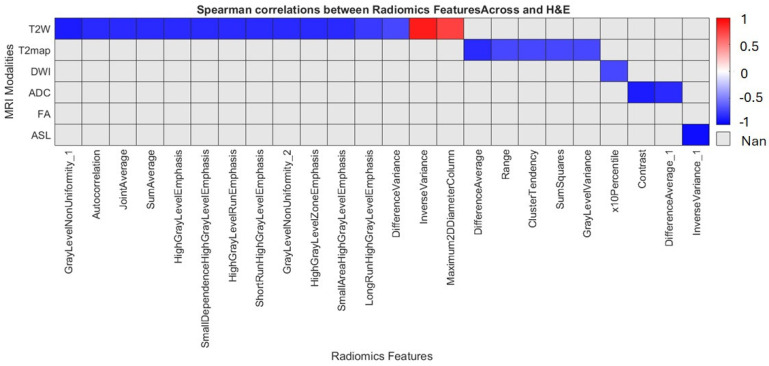
Illustrates the Spearman correlation coefficient of robust MRI radiomic features measured at week 12, in relation to histopathological examination (H&E). Notably, the analysis indicates that the HLA feature does not exhibit any correlation with the radiomic features assessed.

**Figure 7 cancers-17-03545-f007:**
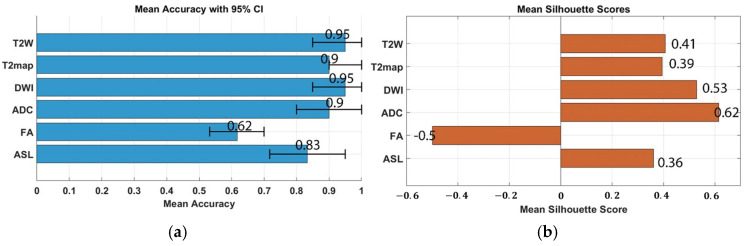
K-means clustering demonstrated measurable (**a**) CV accuracy, with confidence intervals, in distinguishing robust MRI radiomic features within the peritumoral region during weeks 9 to 12 and (**b**) mean silhouette score of k-means.

**Figure 8 cancers-17-03545-f008:**
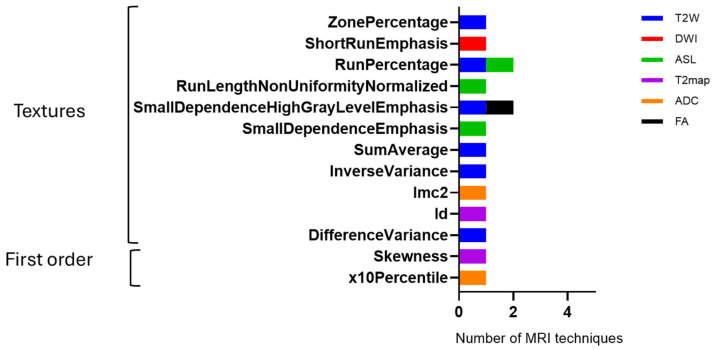
Delta radiomic features across multiple MRI modalities at two time points (week 9 and week 12). Columns represent feature categories (first-order, texture, and shape), while rows correspond to imaging modalities including T2W, DWI, ASL, T2map, ADC, and FA. Notably, several features exhibited changes exceeding 10% between the two time points (week 9 and 12).

**Figure 9 cancers-17-03545-f009:**
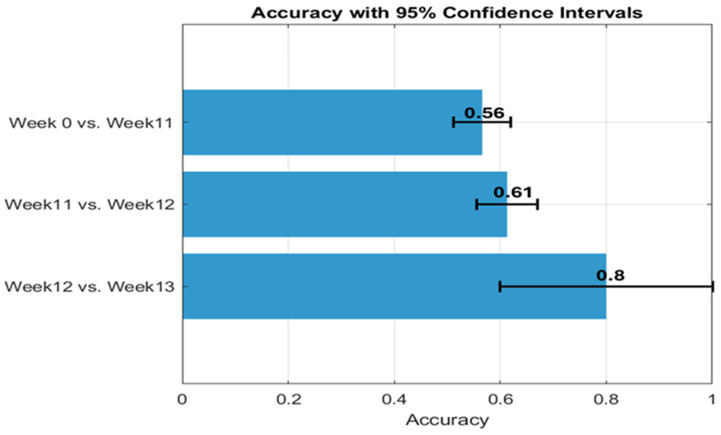
Illustrates the accuracy and 95% confidence interval (CI) of k-means in differentiating between various time points within an external database. This analysis utilizes the top 10 radiomics features extracted from T2W in this study.

**Table 1 cancers-17-03545-t001:** The robust radiomic features exhibit high inter-observer reliability, with Intraclass Correlation Coefficients (ICCs) exceeding 0.8, indicating consistent measurement across different observers. The numerical values represent the ranking of the top radiomic features identified through the mutual information method. * Symbol represents features exhibiting changes exceeding 10% between the two time points. # Symbol represents high Spearman correlation coefficient with H&E.

Categories	Radiomics Features	T2W	T1W	DWI	ASL	T2map	ADC	FA
shape	Maximum2DDiameterColumn	#	X					
firstorder	x10Percentile			#	X	X1	X2 *	
firstorder	x90Percentile			X3		X	X	X1
firstorder	Energy					X2	X	X
firstorder	Entropy					X	X	
firstorder	InterquartileRange					X	X	X
firstorder	Kurtosis				X2	X3		
firstorder	Maximum					X	X	X
firstorder	MeanAbsoluteDeviation					X		
firstorder	Mean			X		X	X	X
firstorder	Median			X	X	X	X	X
firstorder	Minimum		X			X		
firstorder	Range	X9				X #		X
firstorder	RobustMeanAbsoluteDeviation			X2		X		
firstorder	RootMeanSquared			X		X	X	X
firstorder	Skewness			X8	X	X10 *		
firstorder	TotalEnergy					X	X	X
firstorder	Uniformity				X	X7	X7	
firstorder	Variance					X		
glcm	Autocorrelation	X #						X
glcm	ClusterProminence					X		
glcm	ClusterShade					X		
glcm	ClusterTendency					X #		
glcm	Contrast	X				X	X6 #	X
glcm	Correlation			X4			X4	
glcm	DifferenceAverage	X				X #	X10 #	X
glcm	DifferenceEntropy	X	X			X	X	X
glcm	DifferenceVariance	X *	X			X	X	
glcm	Id	X4	X		X4	X8 *	X	X3
glcm	Idm	X	X		X	X	X	X
glcm	Imc1		X			X9	X	X6
glcm	Imc2						X3 *	X
glcm	InverseVariance	X7 *#	X		X6 #	X		X4
glcm	JointAverage	X #						
glcm	JointEnergy				X5			
glcm	MCC							X2
glcm	MaximumProbability				X			
glcm	SumAverage	X *#						
glcm	SumSquares					X #		
gldm	DependenceNonUniformityNormalized	X8	X	X7	X	X	X9	
gldm	DependenceVariance	X6 #	X	X6	X8	X #		
gldm	GrayLevelNonUniformity	X		X	X1			
gldm	GrayLevelVariance					X		
gldm	HighGrayLevelEmphasis	X #						X
gldm	LargeDependenceEmphasis	X	X	X1	X	X4	X1	
gldm	LargeDependenceHighGrayLevelEmphasis	X						X
gldm	LargeDependenceLowGrayLevelEmphasis	X			X3			
gldm	LowGrayLevelEmphasis	X3			X			
gldm	SmallDependenceEmphasis	X	X	X	X *	X	X	
gldm	SmallDependenceHighGrayLevelEmphasis	X *#						X *
gldm	SmallDependenceLowGrayLevelEmphasis	X5			X			
glrlm	GrayLevelNonUniformity	X			X10			
glrlm	GrayLevelNonUniformityNormalized	X			X	X	X	
glrlm	GrayLevelVariance					X		
glrlm	HighGrayLevelRunEmphasis	X #						X
glrlm	LongRunEmphasis	X	X	X	X	X	X	
glrlm	LongRunHighGrayLevelEmphasis	X #						X
glrlm	LongRunLowGrayLevelEmphasis	X			X			
glrlm	LowGrayLevelRunEmphasis	X			X			
glrlm	RunEntropy					X	X	
glrlm	RunLengthNonUniformityNormalized	X	X	X	X	X5	X	
glrlm	RunPercentage	X *	X	X	X *	X	X	
glrlm	RunVariance	X	X	X	X	X	X	
glrlm	ShortRunEmphasis	X	X	X *	X	X	X	
glrlm	ShortRunHighGrayLevelEmphasis	X #						X
glrlm	ShortRunLowGrayLevelEmphasis	X			X7			
glszm	GrayLevelNonUniformity	X2 #			X			
glszm	GrayLevelNonUniformityNormalized					X	X	
glszm	GrayLevelVariance					X		
glszm	HighGrayLevelZoneEmphasis	X #						X
glszm	LargeAreaEmphasis	X	X	X	X	X	X	
glszm	LargeAreaHighGrayLevelEmphasis	X1						X
glszm	LargeAreaLowGrayLevelEmphasis	X			X			
glszm	LowGrayLevelZoneEmphasis	X			X			
glszm	SizeZoneNonUniformityNormalized	X	X	X	X	X	X	
glszm	SmallAreaEmphasis	X	X	X	X	X	X	
glszm	SmallAreaHighGrayLevelEmphasis	X #						X
glszm	SmallAreaLowGrayLevelEmphasis	X			X			X5
glszm	ZoneEntropy						X	
glszm	ZonePercentage	X *	X	X	X9	X	X	
glszm	ZoneVariance	X	X	X5	X	X	X5	
ngtdm	Busyness						X8	
ngtdm	Coarseness	X	X		X	X6	X	
ngtdm	Complexity	X				X		X
ngtdm	Contrast	X						
ngtdm	Strength					X		

## Data Availability

The datasets generated and analyzed during this study, including multiparametric MRI scans, processed radiomics feature matrices, and associated MATLAB scripts, are available in the Zenodo repository at: https://doi.org/10.5281/zenodo.17510028 (accessed on 30 May 2025).

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
