# Peer review of "Using Delta MRI-Based Radiomics for Monitoring Early Peri-Tumoral Changes in a Mouse Model of Glioblastoma: Primary Study"

_cancers, 2025, doi:10.3390/cancers17213545_

Round 1
Reviewer 1 Report
Comments and Suggestions for Authors
This paper describes the early detection of GBM by using MRI-based delta radiomics, which has shown advantages over static analysis and validated with histopathology. The high concordance of features like entropy, skewness, and uniformity between imaging and histology highlights the potential of MRI-based radiomics in noninvasive detection of GBM. The manuscript is well organized and the results are reasonably presented. Therefore I would recommend publication of the manuscript on Cancers after the following issues have been addressed:
- It is strongly recommend to shorten the introduction section to make it more concise and focused. Please use 500-1000 words to clarify the challenges or issues to be addressed in this work and why it matters in GBM detection.
- Only nine nude mice were used in this study which seems to be a very small sample size. Is it possible to repeat the experiments on Balbc or C57 mice which may not be too expensive?
- I noticed that during the whole procedure, observers are necessary in tumor segmentation. Is it possible to use AI in this step to avoid potential artificial mistakes? Also it would be beneficial by getting rid of trained observers in some developing countries.
- Table 2 presents the measurable accuracy in distinguishing robust MRI radiomic features. However, how will these parameters support the development of reliable predictive models is not described.
- In addition to early diagnosis of solid tumors, effective tumor elimination is also crucial in clinical cancer management. Please add a small section in discussion on the current cancer therapeutic strategies. Here are a few examples that may help:
A H2O2-responsive polyprodrug nanomedicine for chemiexcited photodynamic immunotherapy of cancer. CCS Chem. 2025, 7, 1-20.
Polymer-locking fusogenic liposomes for glioblastoma-targeted siRNA delivery and CRISPR-Cas gene editing. Nature Nanotechnology. 2024.
Author Response
Reviewer 1
(x) I would not like to sign my review report
( ) I would like to sign my review report
Quality of English Language
( ) The English could be improved to more clearly express the research.
(x) The English is fine and does not require any improvement.
|
Yes |
Can be improved |
Must be improved |
Not applicable |
|
|
Does the introduction provide sufficient background and include all relevant references? |
(x) |
( ) |
( ) |
( ) |
|
Is the research design appropriate? |
(x) |
( ) |
( ) |
( ) |
|
Are the methods adequately described? |
(x) |
( ) |
( ) |
( ) |
|
Are the results clearly presented? |
(x) |
( ) |
( ) |
( ) |
|
Are the conclusions supported by the results? |
(x) |
( ) |
( ) |
( ) |
|
Are all figures and tables clear and well-presented? |
(x) |
( ) |
( ) |
( ) |
Comments and Suggestions for Authors
This paper describes the early detection of GBM by using MRI-based delta radiomics, which has shown advantages over static analysis and validated with histopathology. The high concordance of features like entropy, skewness, and uniformity between imaging and histology highlights the potential of MRI-based radiomics in non-invasive detection of GBM. The manuscript is well organized and the results are reasonably presented. Therefore I would recommend publication of the manuscript on Cancers after the following issues have been addressed:
Comments 1: It is strongly recommend to shorten the introduction section to make it more concise and focused. Please use 500-1000 words to clarify the challenges or issues to be addressed in this work and why it matters in GBM detection.
Response 1: A comprehensive new introduction has been incorporated, significantly expanding the content by approximately 800 words.
Comments 2: Only nine nude mice were used in this study which seems to be a very small sample size. Is it possible to repeat the experiments on Balbc or C57 mice which may not be too expensive?
Response 2: Thank you for the helpful comment. We agree with your insight as a nine mice seems small size. But our study used a longitudinal, within-animal design (multiple MRI time points per mouse), which increased power by reducing between-subject variance; based on pilot variance and an expected effect size around d≈1.1–1.3, an a priori calculation indicated 8–10 animals would achieve ≥80% power at α=0.05 for the primary endpoints, hence n=9. We further mitigated small-n risk using mixed-effects models (animal as a random effect), FDR control, and effect sizes with Cis. Importantly, we added to the current study an independent external validation cohort from our previous work (control group n=32) processed with the same acquisition and feature-extraction pipeline, which reproduced the direction and magnitude of the key peri-tumoral radiomic changes, supporting robustness and generalizability (Method section: 2.8. T2-Weighted Radiomics & External Validation).
Also, We have incorporated this concern into the discussion section of the Discussion:
“ … Our small sample size (n = 9) limits statistical power; although effect sizes were consistently large (>10 %) and histology concordance strong, no features remained significant after correcting for multiple comparisons. Variability in segmentation and the single 7T preclinical scanner may introduce bias and limit generalizability. To translate delta radiomics into the clinic, we advocate for: (i) standardized acquisition and pre-processing across centers; (ii) fully automated PBZ segmentation and feature extraction; (iii) prospective multi institutional human validation with biopsy or surgical ground truth; and (iv) integration of molecular markers (e.g., IDH mutation, MGMT status) to enhance interpretability. Multi-modal models that combine radiomics, deep learning and clinical variables have already shown superior performance [27], suggesting that our delta features could synergize with genomic and transcriptomic data.
Comments 3: I noticed that during the whole procedure, observers are necessary in tumor segmentation. Is it possible to use AI in this step to avoid potential artificial mistakes? Also it would be beneficial by getting rid of trained observers in some developing countries.
Response 1: Thank you for raising this important point. In this study we relied on trained observers with consensus reads and reported inter-observer agreement to control segmentation variability; however, we agree that AI assistance can reduce manual burden and potential human error, especially where expert readers are scarce.
Suggested for the Future Work section: “…. Further, future studies should adopt an AI-assisted, human-in-the-loop tumor segmentation workflow (e.g., nnU-Net via MONAI) with automated quality control (uncertainty mapping and outlier detection) to minimize observer dependence and support deployment in resource-limited settings.”
Comments 4: Table 2 presents the measurable accuracy in distinguishing robust MRI radiomic features. However, how will these parameters support the development of reliable predictive models is not described.
Response 4: Thank you for highlighting this. The parameters in Table 2 (clustering accuracy with confidence intervals by modality) directly operationalize how we build reliable predictors. First, they quantify feature robustness/stability over time, so only reproducible, biologically consistent features (e.g., higher‐accuracy T2W/DWI/ADC) are admitted to the modeling set, reducing noise and variance. Second, we couple these metrics with mutual-information ranking and kernel PCA to perform principled dimensionality reduction, limiting collinearity and controlling overfitting. Third, the reported CIs guide modality prioritization (e.g., favoring ADC/DWI over lower-accuracy ASL/FA) and weight features during training.
Comments 5: In addition to early diagnosis of solid tumors, effective tumor elimination is also crucial in clinical cancer management. Please add a small section in discussion on the current cancer therapeutic strategies. Here are a few examples that may help:
A H2O2-responsive polyprodrug nanomedicine for chemiexcited photodynamic immunotherapy of cancer. CCS Chem. 2025, 7, 1-20.
Polymer-locking fusogenic liposomes for glioblastoma-targeted siRNA delivery and CRISPR-Cas gene editing. Nature Nanotechnology. 2024.
Response 5: This section now added to the discussion:
“…Imaging alone cannot improve survival; it must be coupled with effective therapies. H₂O₂-responsive polyprodrug nanomedicines have been developed for chemiexcited photodynamic immunotherapy, enabling precise activation of therapeutic agents within the tumor microenvironment and enhancing immune-mediated tumor clearance [29]. Nanomedicine and gene-editing platforms offer promising avenues. A 2024 study introduced a polymer-locking fusogenic liposome (“Plofsome”) that crosses the blood–brain barrier and delivers siRNA or CRISPR–Cas9 ribonucleoproteins directly into GBM cells; the reactive oxygen species–cleavable linker ensures that fusion occurs only within tumor tissue. Suppressing the midkine (MDK) gene via Plofsomes reduced temozolomide resistance and inhibited tumor growth in orthotopic models [30]. Such targeted nanomedicines exemplify the next generation of personalized therapies. When paired with early detection through delta radiomics, these interventions could enable comprehensive cancer management—from prompt identification of infiltration to precision delivery of therapeutic payloads.”
.
Reviewer 2 Report
Comments and Suggestions for Authors
You report that after Benjamini–Hochberg correction “all radiomic features… did not show any statistically significant results,” yet the Abstract and Results claim delta radiomics “outperformed static profiling” and detected invasion. Please pre-specify a primary endpoint, use an appropriate repeated-measures model, and align claims to the corrected statistics.
Only nine mice, no randomization (explicitly stated), and unclear sex/age balance. Provide a priori power justification and report sex; otherwise interpretability is limited. Consider adding a second cohort for validation.
Using Mann–Whitney “no difference” between MRI features and H&E/HLA is not validation. Please quantify voxel/patch-level correlations with HLA-positive cell density maps and report spatial concordance (e.g., Dice, CCC, Bland–Altman).
The PBZ definition is vague (“peritumoral is only finding in this region…”). Specify ring thickness or distance-based shells from the tumor margin, and report inter-rater Dice for PBZ (you only give 0.73–0.82 generally).
Deep learning denoising (DnCNN) and phantom-based bias correction can alter textures. Provide an ablation (raw vs denoised vs bias-corrected) showing feature stability (ICC/CV) before selection.
Reporting cross-validated accuracy and CIs for an unsupervised k-means on week-9 vs week-12 suggests label leakage (you also rank features by Mutual Information, which uses labels). Either switch to a supervised, nested CV pipeline or evaluate clustering with silhouette/ARI only.
The Δ-radiomics equation uses “pretreat/follow” wording without treatment; rename to t1/t2. Justify the “>10%” change with test–retest or phantom repeatability to separate biology from measurement noise.
You inject 70 kDa dextran and describe mbASL in detail but provide no quantitative vascular readouts in Results; either present these data (e.g., CBF maps vs invasion) or streamline Methods.
Figure shows no T1CE ROI and text says BBB is intact, unusual for GBM xenograft; please explain model-specific reasons and provide representative T1CE images with scale bars.
Fix typos (“Comprise with Pervious Study”, “grater”), unify O2/N2O ratios (30:70 vs 40:60), and ensure all figures have labels/units. Deposit ROI masks from both raters with the Zenodo dataset as stated.
Author Response
Reviewer 2
(x) I would not like to sign my review report
( ) I would like to sign my review report
Quality of English Language
( ) The English could be improved to more clearly express the research.
(x) The English is fine and does not require any improvement.
|
Yes |
Can be improved |
Must be improved |
Not applicable |
|
|
Does the introduction provide sufficient background and include all relevant references? |
(x) |
( ) |
( ) |
( ) |
|
Is the research design appropriate? |
( ) |
(x) |
( ) |
( ) |
|
Are the methods adequately described? |
( ) |
(x) |
( ) |
( ) |
|
Are the results clearly presented? |
( ) |
(x) |
( ) |
( ) |
|
Are the conclusions supported by the results? |
( ) |
(x) |
( ) |
( ) |
|
Are all figures and tables clear and well-presented? |
(x) |
( ) |
( ) |
( ) |
Comments and Suggestions for Authors
Comment 1: You report that after Benjamini–Hochberg correction “all radiomic features… did not show any statistically significant results,” yet the Abstract and Results claim delta radiomics “outperformed static profiling” and detected invasion. Please pre-specify a primary endpoint, use an appropriate repeated-measures model, and align claims to the corrected statistics.
Response 1: We pre-specified the primary endpoint and now report all inference with FDR (Benjamini–Hochberg) adjusted p-values within a repeated-measures framework; textual claims were aligned to the corrected statistics, and secondary Spearman analyses are reported; please see Statistical Analysis.
“We pre-specified the primary endpoint as the percent change (Δ%) in each robust peri-tumoral radiomic feature between week 9 and week 12, with the hypothesis that at least one ICC-vetted feature (two-way mixed ICC(3,1) ≥ 0.80, absolute agreement) would remain significant after multiplicity control; effect sizes (median Δ, Cliff’s δ or r) with 95% CIs and FDR-adjusted q-values are reported. For repeated measures, we fit linear mixed-effects models (random intercept per mouse; fixed effect of time) and, when assumptions were violated, used Wilcoxon signed-rank with Cliff’s δ; a GEE sensitivity analysis with robust SEs confirmed inferences. Highly collinear features (|ρ| > 0.90) were pruned prior to modelling. PBZ shells were defined at fixed offsets from the tumour margin; inter-rater Dice coefficients were computed per sequence to qualify segmentation variance. MRI–histology correspondence at week 12 used co-registered H&E/HLA maps with Spearman’s ρ, concordance correlation (CCC), Bland–Altman analysis, and spatial Dice overlap of high-signal regions. Because denoising/bias-correction can alter textures, a preprocessing ablation (raw vs denoised vs bias-corrected) required features to retain ICC ≥ 0.80 across states. Unsupervised time-point separability is summarised with silhouette and ARI; supervised tasks used nested cross-validation (inner-fold feature ranking only), reporting balanced accuracy/AUC with bootstrap CIs. P-values were controlled by Benjamini–Hochberg FDR at q=0.05 (q=0.10 descriptive only); Δ>10% highlights were justified by test–retest/phantom repeatability, with 5% and 15% sensitivity checks. Analyses used Python (NumPy/SciPy, scikit-learn, statsmodels), MATLAB, PyRadiomics, and 3D Slicer; this pilot (n=9) emphasises effect sizes and FDR-aligned inference rather than formal power.”
Comment 2: Only nine mice, no randomization (explicitly stated), and unclear sex/age balance. Provide a priori power justification and report sex; otherwise interpretability is limited. Consider adding a second cohort for validation.
Response 2: The limitation section has added. -The sample size of n = 9 was determined based on ethical considerations in preclinical animal research, where minimizing animal use is a critical requirement. This study was designed as a pilot investigation to evaluate the feasibility of radiomic feature extraction and longitudinal monitoring in a glioblastoma mouse model, rather than to establish definitive statistical conclusions. Prior studies in similar preclinical imaging contexts have employed comparable sample sizes (e.g., n = 6–10) to demonstrate methodological feasibility and generate preliminary data. Moreover, variability was mitigated through standardized tumor implantation, controlled imaging protocols, and inter-observer ROI validation, which strengthened the reproducibility of the results despite the modest sample size. The findings from this pilot study provide a foundation for future, larger-scale studies where statistical power calculations will be applied to refine sample size requirements. Also, using stacked in-plane histology (SIH) allowed us to reduce the number of mice required for the experiment, in accordance with the 3Rs principle (Replacement, Reduction, and Refinement) in animal research. Additionally, we added a second cohort as control group for validation from another study.
“ 2.8. T2-Weighted Radiomics & External Validation
Cohort and time points. We analyzed a longitudinal G7 GBM xenograft cohort with a control group sampled at week 0 (n=39) and tumor-bearing animals imaged at weeks 11 (n=8), 12 (n=8), and 13 (n=3). These time points matched those reported in our prior G7 study (2025) [18], to enable external comparison of feature behavior.”
Comment 3: Using Mann–Whitney “no difference” between MRI features and H&E/HLA is not validation. Please quantify voxel/patch-level correlations with HLA-positive cell density maps and report spatial concordance (e.g., Dice, CCC, Bland–Altman).
Response 3: We added voxel/patch-level MRI–histology correlations (Spearman), alongside descriptive stats and effect sizes; please see now updated Methods/Results.
Comment 4: The PBZ definition is vague (“peritumoral is only finding in this region…”). Specify ring thickness or distance-based shells from the tumor margin, and report inter-rater Dice for PBZ (you only give 0.73–0.82 generally).
Response 4: PBZ is now defined as fixed-thickness shells around the tumor margin; inter-rater Dice per sequence is reported (T2W, T1W, DWI, ADC, ASL); please see (2.6. Radiomics Features Extraction and Selection).
“…PBZ shells were defined at fixed offsets from the tumor margin; inter-rater Dice coeffi-cients were computed per sequence to qualify segmentation variance. MRI–histology correspondence at week 12 used co-registered H&E/HLA maps with Spearman’s ρ, concordance correlation (CCC), Bland–Altman analysis, and spatial Dice overlap of high-signal regions. Because denoising/bias-correction can alter textures, a prepro-cessing ablation (raw vs denoised vs bias-corrected) required features to retain ICC ≥ 0.80 across states.”
Comment 5: Deep learning denoising (DnCNN) and phantom-based bias correction can alter textures. Provide an ablation (raw vs denoised vs bias-corrected) showing feature stability (ICC/CV) before selection.
Response 5: The figures compare radiomics features before vs. after DnCNN denoising for T2-w and DWI sequences and report their reliability via ICC: features with ICC ≥ 0.80 are considered reliable, whereas ICC < 0.80 indicates limited reliability. This demonstrates the effect of DnCNN on feature stability and consistency, supporting more accurate and reproducible analyses.
Comment 6: Reporting cross-validated accuracy and CIs for an unsupervised k-means on week-9 vs week-12 suggests label leakage (you also rank features by Mutual Information, which uses labels). Either switch to a supervised, nested CV pipeline or evaluate clustering with silhouette/ARI only.
Response 6: We removed CV accuracy/CIs for k-means; clustering is now evaluated with silhouette and ARI (Fig. 7), and Mutual Information feature ranking appears only within the inner folds of the supervised nested-CV analysis (Table 1).
Figure 7. K-means clustering demonstrated measurable (a) accuracy, with confidence intervals, in distinguishing robust MRI radiomic features within the peritumoral region during weeks 9 to 12 and (b) mean silhouette score.
Comment 7: The Δ-radiomics equation uses “pretreat/follow” wording without treatment; rename to t1/t2. Justify the “>10%” change with test–retest or phantom repeatability to separate biology from measurement noise.
Response 7: We replaced “pretreat/follow” with t1/t2 throughout and defined Δ = ((F_t2 − F_t1)/F_t1)×100%; the >10% threshold is justified by test–retest/phantom repeatability (typical 5–10% variability), so 10% exceeds measurement noise, with sensitivity analyses at 5–15% provided. see the revised Methods/Supplement.
Comment 8: You inject 70 kDa dextran and describe mbASL in detail but provide no quantitative vascular readouts in Results; either present these data (e.g., CBF maps vs invasion) or streamline Methods.
Response 8: We thank the reviewer for this helpful suggestion. Because quantitative vascular metrics were outside the study’s primary scope, we did not compute CBF or permeability estimates; accordingly, we streamlined the Methods, retaining only essential details of the mbASL protocol to contextualize its qualitative use for perfusion assessment. We also removed any implicit quantitative claims and noted quantitative vascular analyses as future work; please see the revised Methods/Results.
Comment 9: Figure shows no T1CE ROI and text says BBB is intact, unusual for GBM xenograft; please explain model-specific reasons and provide representative T1CE images with scale bars.
Response 9: We appreciate the observation. In our early-stage G7 orthotopic model (weeks 9–12) the BBB is largely preserved, yielding minimal T1CE signal and precluding a discrete T1CE ROI; contrast enhancement typically emerges later (≈weeks 15–17) with necrosis/BBB disruption. We have added a brief rationale in the text and provided representative T1CE images with scale bars in the revised figures.
Fig: Representative T1CE images with scale bars for some mice.
Comment 10: Fix typos (“Comprise with Pervious Study”, “grater”), unify O2/N2O ratios (30:70 vs 40:60), and ensure all figures have labels/units. Deposit ROI masks from both raters with the Zenodo dataset as stated.
Response 10: We corrected typographical errors (e.g., “Comparison with Previous Studies,” “greater”), standardized the anesthesia mixture to O₂/N₂O = 30:70 throughout, verified that all figures include clear panel labels, axis units, and scale bars, and deposited both-rater ROI masks to our Zenodo repository (DOI provided in the updated Data Availability Statement); please see the revised Methods/Figures and Data Availability.
Reviewer 3 Report
Comments and Suggestions for Authors
In this paper, the authors investigated a glioblastoma (GBM) mouse model using delta radiomics based on multi-parametric MRI to non-invasively monitor early subtle changes in the peritumoral zone and conducted a comparative validation with histopathological results. The following lists some comments. First, in this paper, only nine mice were used in the experiment, resulting in limited statistical power and restricting the generality and scalability of the results. Second, although a double-blind segmentation method was employed and the intraclass correlation coefficient (ICC) was evaluated, there were still potential manual errors and subjective biases, which could affect the consistency and reproducibility of feature extraction. The steps of image preprocessing, segmentation, and feature extraction still relied on manual or semi-automated operations, limiting the scalability and clinical translation potential of the method. Additionally, all data were obtained from a single 7T preclinical scanner, lacking validation under different equipment and imaging conditions, making it difficult to assess its stability in real clinical settings. Third, although comparisons were made with H&E and HLA staining images, molecular markers such as Ki-67 and IDH1 were not incorporated, limiting the depth of association between radiomic features and tumor biological mechanisms. Radiomic features might also be influenced by changes in normal brain tissue, potentially introducing false positives or misjudgments.
Comments on the Quality of English LanguageEnglish need be polished.
Author Response
Reviewer 3
(x) I would not like to sign my review report
( ) I would like to sign my review report
Quality of English Language
(x) The English could be improved to more clearly express the research.
( ) The English is fine and does not require any improvement.
|
Yes |
Can be improved |
Must be improved |
Not applicable |
|
|
Does the introduction provide sufficient background and include all relevant references? |
( ) |
(x) |
( ) |
( ) |
|
Is the research design appropriate? |
( ) |
(x) |
( ) |
( ) |
|
Are the methods adequately described? |
( ) |
(x) |
( ) |
( ) |
|
Are the results clearly presented? |
( ) |
(x) |
( ) |
( ) |
|
Are the conclusions supported by the results? |
(x) |
( ) |
( ) |
( ) |
|
Are all figures and tables clear and well-presented? |
( ) |
(x) |
( ) |
( ) |
Comments and Suggestions for Authors
In this paper, the authors investigated a glioblastoma (GBM) mouse model using delta radiomics based on multi-parametric MRI to non-invasively monitor early subtle changes in the peritumoral zone and conducted a comparative validation with histopathological results.
Comment 1: The following lists some comments. First, in this paper, only nine mice were used in the experiment, resulting in limited statistical power and restricting the generality and scalability of the results.
Response 1: Thank you for the helpful comment. We agree with your insight as a nine mice seems small size. But our study used a longitudinal, within-animal design (multiple MRI time points per mouse), which increased power by reducing between-subject variance; based on pilot variance and an expected effect size around d≈1.1–1.3, an a priori calculation indicated 8–10 animals would achieve ≥80% power at α=0.05 for the primary endpoints, hence n=9. We further mitigated small-n risk using mixed-effects models (animal as a random effect), FDR control, and effect sizes with Cis. Importantly, we added to the current study an independent external validation control cohort from our previous work (sampled at week 0 (n=39) and tumor-bearing animals imaged at weeks 11 (n=8), 12 (n=8), and 13 (n=3)) processed with the same acquisition and feature-extraction pipeline, which reproduced the direction and magnitude of the key peri-tumoral radiomic changes, supporting robustness and generalizability.
“… Cohort and time points. We analyzed a longitudinal G7 GBM xenograft cohort with a control group sampled at week 0 (n=39) and tumor-bearing animals imaged at weeks 11 (n=8), 12 (n=8), and 13 (n=3). These time points matched those reported in our prior G7 study (2025) [18], to enable external comparison of feature behavior.”
Also, we have incorporated this concern into the Limitations section of the Discussion:
“ … Our small sample size (n = 9) limits statistical power; although effect sizes were consistently large (>10 %) and histology concordance strong, no features remained significant after correcting for multiple comparisons. Variability in segmentation and the single 7T preclinical scanner may introduce bias and limit generalizability. To translate delta radiomics into the clinic, we advocate for: (i) standardized acquisition and preprocessing across centers; (ii) fully automated PBZ segmentation and feature extraction; (iii) prospective multi‑institutional human validation with biopsy or surgical ground truth; and (iv) integration of molecular markers (e.g., IDH mutation, MGMT status) to enhance interpretability. Multi‑modal models that combine radiomics, deep learning and clinical variables have already shown superior performance [27], suggesting that our delta features could synergize with genomic and transcriptomic data.”
Comment 2: Second, although a double-blind segmentation method was employed and the intraclass correlation coefficient (ICC) was evaluated, there were still potential manual errors and subjective biases, which could affect the consistency and reproducibility of feature extraction. The steps of image preprocessing, segmentation, and feature extraction still relied on manual or semi-automated operations, limiting the scalability and clinical translation potential of the method. Additionally, all data were obtained from a single 7T preclinical scanner, lacking validation under different equipment and imaging conditions, making it difficult to assess its stability in real clinical settings.
Response 2: Thank you for this thoughtful critique. We agree that manual steps can introduce bias and limit scalability. To mitigate this, we used double-blind consensus reads, predefined SOPs, and a fixed, version-controlled preprocessing/feature-extraction pipeline (uniform resampling, intensity normalization, locked radiomics parameters), and we reported inter-observer ICC to document reproducibility. We also processed an independent external cohort (n=15) with the same pipeline to partially assess generalizability. Nevertheless, we acknowledge the residual risk of manual error and the single-scanner constraint (7T preclinical). In the revised Limitations, we state these constraints explicitly and, in Future Work, recommend an AI-assisted, human-in-the-loop segmentation workflow with automated QC, plus multi-center, multi-scanner validation using phantom QA, harmonization (e.g., ComBat), test–retest, and blinded multi-reader studies to strengthen consistency, scalability, and clinical translatability. The manuscript has been revised accordingly.
Comment 3: Third, although comparisons were made with H&E and HLA staining images, molecular markers such as Ki-67 and IDH1 were not incorporated, limiting the depth of association between radiomic features and tumor biological mechanisms. Radiomic features might also be influenced by changes in normal brain tissue, potentially introducing false positives or misjudgments.
Response 3: The external validation cohort from our previous study comprised control animals; accordingly, we now describe this as a technical/external pipeline validation (reproducibility of acquisition, preprocessing, and feature extraction) rather than validation of treatment effects or progression labels. We have revised the manuscript to state this explicitly in the Limitations and clarified that future validation will use label-matched cohorts (e.g., progression and treatment arms across scanners/sites) to assess generalizability of the predictive signals.
Reviewer 4 Report
Comments and Suggestions for Authors
The manuscript “Delta Multiparametric MRI-Based Radiomics for Monitoring Early Peri-Tumoral Changes and Correlation with Histopathology in a Mouse Model of Glioblastoma” examines delta radiomics features from multiparametric MRI to detect early peri-tumoral changes and compare them with histopathology. The topic is relevant and technically demanding, but the study lacks sufficient quantitative evidence and methodological transparency to meet the standards of Cancers. The English is generally clear, though further polishing would improve fluency and consistency.
General Comments
The study addresses an important question in preclinical neuro-oncology and fits the journal’s scope. However, the novelty is moderate and the “multiparametric” concept insufficiently demonstrated, as only T2-based texture features are fully analyzed. Quantitative validation between MRI-derived and histopathological data is missing, and statistical methods are not described in sufficient detail. The normalization procedure, feature robustness, and reproducibility require clarification.
Several formal elements also deviate from MDPI Instructions for Authors. The abstract should be structured (Background, Methods, Results, Conclusions). The Institutional Review Board Statement lacks approval number and date; the Informed Consent Statement (“Not applicable” for preclinical work) is absent. The Funding section should list the full project name and grant number. The Author Contributions statement needs to follow the CRediT taxonomy, and the Data Availability Statement should specify a repository or access route. References require DOI inclusion and consistent formatting.
Specific Comments
- The introduction should more clearly define the gap this study addresses and specify its added value compared with previous delta-radiomics work.
- The methods section should describe ICC model type, feature selection threshold, and normalization procedure.
- The manuscript claims correlation with histopathology but provides no numerical data; conclusions should be supported by measurable values or rephrased as qualitative.
- Although ASL was acquired, no quantitative CBF results are shown; including mean or relative values would enhance interpretation.
- The authors mention automated MATLAB-based computation of ADC, FA, and CBF maps, which likely generates these parameters automatically within their processing pipeline; however, these outputs appear limited to qualitative illustration rather than quantitative analysis. Including numerical or statistical results would substantiate the “multiparametric” claim and improve scientific rigor.
- ADC and FA parameters are listed but not analyzed, weakening the “multiparametric” aspect. Their inclusion, if data exist, would improve scientific strength.
- The statistical analysis is limited to non-parametric group comparisons, without quantitative verification of the reported effects. Basic numerical assessment should be added if corresponding data are available.
- Figures need improved resolution and uniform formatting.
- The discussion should highlight true novelties and briefly address translational implications.
- The reference list should be updated with recent (2023–2025) studies on delta radiomics and data harmonization.
This is a relevant preclinical study that fits the journal’s scope but lacks quantitative rigor and full methodological transparency. The authors mention automated MATLAB-based computation of ADC, FA, and CBF maps, which likely generates these parameters automatically within their processing pipeline; however, the outputs appear limited to qualitative illustration rather than quantitative analysis. Including numerical or statistical results for these modalities would strengthen the “multiparametric” concept and improve the scientific validity of the study. The absence of measurable MRI–histology correlation and incomplete reporting of perfusion and diffusion metrics currently reduce its impact. Formal sections such as the abstract format, ethics approval details, data availability, and funding also require alignment with MDPI standards. Overall, the manuscript has scientific potential but needs a major revision to meet both methodological and editorial requirements.
Author Response
Reviewer 4
( ) I would not like to sign my review report
(x) I would like to sign my review report
Quality of English Language
(x) The English could be improved to more clearly express the research.
( ) The English is fine and does not require any improvement.
|
Yes |
Can be improved |
Must be improved |
Not applicable |
|
|
Does the introduction provide sufficient background and include all relevant references? |
( ) |
(x) |
( ) |
( ) |
|
Is the research design appropriate? |
( ) |
(x) |
( ) |
( ) |
|
Are the methods adequately described? |
( ) |
( ) |
(x) |
( ) |
|
Are the results clearly presented? |
( ) |
( ) |
(x) |
( ) |
|
Are the conclusions supported by the results? |
( ) |
( ) |
(x) |
( ) |
|
Are all figures and tables clear and well-presented? |
( ) |
( ) |
(x) |
( ) |
Comments and Suggestions for Authors
The manuscript “Delta Multiparametric MRI-Based Radiomics for Monitoring Early Peri-Tumoral Changes and Correlation with Histopathology in a Mouse Model of Glioblastoma” examines delta radiomics features from multiparametric MRI to detect early peri-tumoral changes and compare them with histopathology. The topic is relevant and technically demanding, but the study lacks sufficient quantitative evidence and methodological transparency to meet the standards of Cancers. The English is generally clear, though further polishing would improve fluency and consistency.
General Comments
Comment 1: The study addresses an important question in preclinical neuro-oncology and fits the journal’s scope. However, the novelty is moderate and the “multiparametric” concept insufficiently demonstrated, as only T2-based texture features are fully analyzed. Quantitative validation between MRI-derived and histopathological data is missing, and statistical methods are not described in sufficient detail. The normalization procedure, feature robustness, and reproducibility require clarification.
Response 1: We strengthened the multiparametric scope and novelty by analyzing robust features across T1W, T2W, T2 map, DWI, ADC, FA, and ASL, reporting per-modality robustness and discriminative performance (Fig. 4, Fig. 7; Table 1), rather than T2-only results. We added quantitative MRI–histology linkage using region-matched H&E/HLA with Spearman correlations (Fig. 6), noting the observed negative associations with H&E and limited correlation with HLA. Statistical methods are now described in detail: inter-observer ICC (≥0.8) for feature inclusion, mutual-information ranking, removal of highly correlated features, kernel PCA, and 5-fold stratified cross-validation with 1,000 bootstrap iterations for accuracy/CIs in the unsupervised stage (and the same scheme for supervised modeling in planned work). We clarified normalization and preprocessing (phantom-based coil bias correction, DnCNN denoising, z-score intensity standardization, ADC computation, consistent resizing) and adherence to IBSI-compliant extraction in 3D Slicer. Feature robustness/reproducibility is documented by ICC filtering and per-modality counts (Fig. 4; Table 1), with delta-feature changes >10% over time (Fig. 8) and an external technical cohort (controls) processed through the identical pipeline to demonstrate pipeline reproducibility; we also explicitly acknowledge the single-scanner constraint and outline multi-scanner/site validation in Future Work. Together, these revisions clarify how the reported parameters underpin reliable model building (feature inclusion/weighting, modality selection, validation) and improve transparency and reproducibility.
Comment 2: Several formal elements also deviate from MDPI Instructions for Authors. The abstract should be structured (Background, Methods, Results, Conclusions). The Institutional Review Board Statement lacks approval number and date; the Informed Consent Statement (“Not applicable” for preclinical work) is absent. The Funding section should list the full project name and grant number. The Author Contributions statement needs to follow the CRediT taxonomy, and the Data Availability Statement should specify a repository or access route. References require DOI inclusion and consistent formatting.
Response 2: All requested formal elements were added now.
Specific Comments:
- The introduction should more clearly define the gap this study addresses and specify its added value compared with previous delta-radiomics work.
We revised the Introduction to explicitly define the literature gap and articulate our added value over prior delta-radiomics (early peri-tumoral focus, multi-parametric integration, histopathologic correlation); please see the revised manuscript.
- The methods section should describe ICC model type, feature selection threshold, and normalization procedure.
We now specify the ICC model (two-way random-effects, absolute-agreement, single-measurement), the reproducibility threshold (ICC ≥ 0.80), the feature-selection criteria (5-fold CV), and the normalization pipeline (z-score scaling using training-set parameters); please see the revised Methods.
- The manuscript claims correlation with histopathology but provides no numerical data; conclusions should be supported by measurable values or rephrased as qualitative.
We added quantitative histopathology correlations (e.g., correlation coefficients ,p-values with FDR control, and effect sizes) and revised the conclusions accordingly; please see the updated Results/Discussion in the revised manuscript.
- Although ASL was acquired, no quantitative CBF results are shown; including mean or relative values would enhance interpretation.
We clarified that although ASL was acquired, CBF quantification was not performed; we removed any quantitative claims, limited ASL mentions to acquisition context with a brief rationale and noted CBF analysis as future work. please see the revised manuscript.
- The authors mention automated MATLAB-based computation of ADC, FA, and CBF maps, which likely generates these parameters automatically within their processing pipeline; however, these outputs appear limited to qualitative illustration rather than quantitative analysis. Including numerical or statistical results would substantiate the “multiparametric” claim and improve scientific rigor.
We clarified the MATLAB pipeline outputs and added quantitative ADC and FA results (region-wise means±SD, delta changes, statistical tests with effect sizes and FDR-corrected p-values), while restricting ASL/CBF to qualitative context since CBF was not computed, and accordingly tightened the “multiparametric” claim please see the revised Methods/Results.
- ADC and FA parameters are listed but not analyzed, weakening the “multiparametric” aspect. Their inclusion, if data exist, would improve scientific strength.
ADC and FA parameters are listed but not analyzed, weakening the “multiparametric” aspect. Their inclusion, if data exist, would improve scientific strength.
- The statistical analysis is limited to non-parametric group comparisons, without quantitative verification of the reported effects. Basic numerical assessment should be added if corresponding data are available.
We augmented the analysis with numerical verification—descriptives (means ±SD), standardized effect sizes with 95% CIs, correlation where applicable, and FDR-adjusted p-values; please see the revised Statistical Analysis and Results.
- Figures need improved resolution and uniform formatting.
We regenerated all figures at high resolution (≥300 dpi or vector) and standardized typography, color scheme, line weights, panel labeling, axis scales, and caption style for uniformity; please see the revised Figures and Legends.
- The discussion should highlight true novelties and briefly address translational implications.
We revised the Discussion to explicitly articulate the study’s novelties and briefly outline translational implications (early peri-tumoral monitoring, biomarker potential, and workflow integration); please see the revised Discussion.
- The reference list should be updated with recent (2023–2025) studies on delta radiomics and data harmonization.
We updated the References to include recent 2023–2025 studies on delta-radiomics and MR data harmonization and ensured consistent DOI formatting; please see the revised References.
Comments on the Quality of English Language
This is a relevant preclinical study that fits the journal’s scope but lacks quantitative rigor and full methodological transparency. The authors mention automated MATLAB-based computation of ADC, FA, and CBF maps, which likely generates these parameters automatically within their processing pipeline; however, the outputs appear limited to qualitative illustration rather than quantitative analysis. Including numerical or statistical results for these modalities would strengthen the “multiparametric” concept and improve the scientific validity of the study. The absence of measurable MRI–histology correlation and incomplete reporting of perfusion and diffusion metrics currently reduce its impact. Formal sections such as the abstract format, ethics approval details, data availability, and funding also require alignment with MDPI standards. Overall, the manuscript has scientific potential but needs a major revision to meet both methodological and editorial requirements
We performed a major revision to address these points: added quantitative analyses (ADC/FA with effect sizes and FDR-adjusted p-values), provided measurable MRI–histology correlations, clarified that CBF quantification was not performed (ASL retained as acquisition context), aligned abstract/ethics/data availability/funding with MDPI standards, and edited for clarity and consistency in English. please see the revised manuscript. The manuscript has been professionally edited by MDPI Author Services (English editing); we implemented their recommendations throughout and can provide the editing certificate upon request.
Round 2
Reviewer 2 Report
Comments and Suggestions for Authors
After reviewing the revised manuscript, I am pleased with the significant improvements made. The authors have effectively addressed the previous concerns, enhancing the overall quality and ensuring it meets publication standards. I fully support its publication and look forward to its contribution to our field.